# The molecular basis for SARS-CoV-2 binding to dog ACE2

Zengyuan Zhang[1,2,9], Yanfang Zhang[1,3,9], Kefang Liu[1,9], Yan Li[1,9], Qiong Lu[4,9], Qingling Wang[5,9], Yuqin Zhang[1,2], Liang Wang [1], Hanyi Liao[1,2], Anqi Zheng[1,2], Sufang Ma[1], Zheng Fan[6], Huifang Li[7], Weijin Huang [4], Yuhai Bi [1], Xin Zhao [1], Qihui Wang [1], George F. Gao [1✉], Haixia Xiao [3✉], Zhou Tong [1✉], Jianxun Qi [1,8✉] & Yeping Sun [1✉]

SARS-CoV-2 can infect many domestic animals, including dogs. Herein, we show that dog angiotensin-converting enzyme 2 (dACE2) can bind to the SARS-CoV-2 spike (S) protein receptor binding domain (RBD), and that both pseudotyped and authentic SARS-CoV-2 can infect dACE2-expressing cells. We solved the crystal structure of RBD in complex with dACE2 and found that the total number of contact residues, contact atoms, hydrogen bonds and salt bridges at the binding interface in this complex are slightly fewer than those in the complex of the RBD and human ACE2 (hACE2). This result is consistent with the fact that the binding affinity of RBD to dACE2 is lower than that of hACE2. We further show that a few important mutations in the RBD binding interface play a pivotal role in the binding affinity of RBD to both dACE2 and hACE2. Our work reveals a molecular basis for cross-species transmission and potential animal spread of SARS-CoV-2, and provides new clues to block the potential transmission chains of this virus.

[1] CAS Key Laboratory of Pathogenic Microbiology and Immunology, Institute of Microbiology, Chinese Academy of Sciences, Beijing, China. [2] University of Chinese Academy of Sciences, Beijing, China. [3] Tianjin Institute of Industrial Biotechnology, Chinese Academy of Sciences, Tianjin, China. [4] Division of HIV/AIDS and Sex-Transmitted Virus Vaccines, National Institutes for Food and Drug Control (NIFDC), Beijing, China. [5] Shanxi Natural Carbohydrate Resource Engineering Research Center, College of Food Science and Technology, Northwest University, Xi'an, China. [6] Institute of Microbiology, Chinese Academy of Sciences, Beijing, China. [7] The Northern Medical District of the PLA General Hospital, Beijing, China. [8] Savaid Medical School, University of Chinese Academy of Sciences, Beijing, China. [9] These authors contributed equally: Zengyuan Zhang, Yanfang Zhang, Kefang Liu, Yan Li, Qiong Lu, Qingling Wang. ✉email: gaof@im.ac.cn; xiao_hx@tib.cas.cn; tongz@im.ac.cn; jxqi@im.ac.cn; sunyeping@im.ac.cn

There is a continuously escalating threat from emerging and re-emerging viral diseases on human health[1]. The ongoing coronavirus disease 2019 (COVID-19) pandemic, caused by the severe acute respiratory syndrome coronavirus 2 (SARS-CoV-2), highlights the serious challenges faced by global public health. As of June 21, 2021, the number of confirmed COVID-19 cases have accumulated to more than 178.5 million, including over 3.8 million deaths[2].

One possible origin of COVID-19 is the cross-species transmission of SARS-CoV-2 from animals to humans. SARS-CoV-2 shares a whole-genome identity of 96% with a bat-derived CoV, BatCoV RaTG13, from *Rhinolophus affinis* in Yunnan Province, China[3]. In addition to BatCoV RaTG13, another SARS-CoV-2-like CoV, sharing 91.02% of the whole-genome identity to SARS-CoV-2, was isolated from dead Malayan pangolins[4]. However, the S protein of neither BatCoV RaTG13 nor the pangolin SARS-CoV-2-like CoV harbors the functional polybasic (furin) cleavage site at the S1–S2 boundary found in the S protein of SARS-CoV-2, which suggests that the virus may not directly jump from these two species to humans. An observation of an intermediate or fully formed polybasic cleavage site in a SARS-CoV-2-like virus from animals would lend further support to identify the direct origin of SARS-CoV-2[5].

SARS-CoV-2 may have a wide range of hosts. Some domestic animals, including several pets that are in close contact with humans, are susceptible to the virus. For example, the virus was shown to replicate efficiently in ferrets and cats, but poorly in dogs[6]. A recent study provided evidence that angiotensin-converting enzyme 2 (ACE2) from many animals, including Primates (monkey), *Lagomorpha* (rabbit), *Pholidota* (Malayan pangolin), *Perissodactyla* (horse), most Carnivora (cat, fox, dog, and raccoon dog), and most Artiodactyla (pig, wild Bactrian camel, bovine, goat, and sheep) can bind to the SARS-CoV-2 S protein receptor binding domain (RBD), similar to their human counterpart, human ACE2 (hACE2)[7]. hACE2 is the human receptor of SARS-CoV-2[8–10], so SARS-CoV-2 may also exploit ACE2 from other animals as a receptor to infect these animals.

Dogs are one of the most popular domestic pets worldwide and are in a close contact with humans. They could play a role in SARS-CoV-2 transmission if they are infected. Indeed, dogs from households with confirmed human cases of COVID-19 in Hong Kong were confirmed to be infected with SARS-CoV-2, and the genetic sequences of the viruses from two dogs were identical to that of the viruses detected in the respective human cases. Evidence suggests that these are instances of human-to-animal transmission of SARS-CoV-2[11]. Recently, a large-scale epidemiological survey that included 919 cats and dogs living in Italy showed that 3.3% of dogs and 5.8% of cats were SARS-CoV-2 neutralizing antibody positive, and dogs from COVID-19 positive households were significantly more likely to test positive than those from COVID-19-negative households[12]. These results further stress the potential risk of these pets in the spread of SARS-CoV-2.

To define the variance at SARS-COV-2 infection efficacies between dog ACE2 (dACE2)- and hACE2-overexpressed stable cell lines, as well as to determine the interaction difference between RBD-dACE2 and RBD-hACE2, we tested the binding affinity of RBD and RBD mutants to dACE2 and hACE2 by biochemical approaches and further solved the crystal structure of RBD in complex with dACE2. Furthermore, we analyzed the residues of RBD that are involved in the interaction with either dACE2 or hACE2. Based on the results and accumulated evidence, we demonstrated that a few important mutations in the RBD-binding interface play a pivotal role in the binding affinity to dACE2 and hACE2. Our work provides insight for cross-species transmission and potential animal spread of SARS-CoV-2,

and highlights the importance of closely monitoring the related virus mutations at the human-animal interface.

## Results

### The binding affinity of dACE2/hACE2 to RBD and infectivity of pseudotyped and authentic viruses. SARS-CoV-2 S glycoprotein is a protein of 1273 residues. It harbors a furin cleavage site (Q677TNSPRRAR↓SV687) at the boundary between the S1/S2 subunits[13]. The S1 domain contains two subdomains: the N-terminal domain and the C-terminal domain (CTD). RBD is responsible for receptor recognition, which had been mapped to the CTD in previous structural studies[9,10]. dACE2 shares 83.88% primary sequence identity with hACE2. It is also composed of two subdomains, subdomains I and II (Fig. 1a). Because of the high sequence consensus between hACE2 and dACE2, we speculated that dACE2 may also be able to bind to RBD. Therefore, we determined the binding affinities of RBD to both hACE2 and dACE2 using surface plasmon resonance (SPR). The results showed that the dissociation constant ($K_D$) between RBD and hACE2 was 18.5 nM, while that between RBD and dACE2 was 123 nM, which confirms that dACE2 can indeed bind to RBD, but with a binding affinity 6.65 times lower than that of hACE2 (Fig. 1b and c).

To test the hypothesis that dACE2 is a receptor for SARS-CoV-2, we infected dACE2-transfected BHK21 cells with a pseudovirus bearing SARS-CoV-2 S protein. Our results showed that the fluorescence signal represented as the relative luminescence units (RLU) in the S protein-expressing BHK21 cells had a dose-dependence relationship with the virus dilutions. At virus dilutions of 60 and 180, the RLU values in the dACE2-expressing BHK21 cells were significantly higher than those in the BHK21 cells without expressing dACE2 ($p < 0.0001$, Student's *t*-test), but at the virus dilutions lower than 180, there was no significant difference between the RLU values of BHK21 cells expressing dACE2 and those not expressing dACE2 (Fig. 1d). In contrast, the SARS-CoV-2 S protein-bearing pseudovirus infection led to significantly higher RLU values in the hACE2-expressing BHK21 cells than those not expressing hACE2 at all dilutions (Fig. 1e). Similarly, SARS-CoV-2 S protein-bearing pseudovirus infection produced significantly higher RLU values in hACE2-expressing HeLa cells when the virus dilution was above 540, and in dACE2-expressing Hella cells when the virus dilution was above 1620 than those not expressing the two ACE2 molecules, respectively (Supplementary Fig. 1). These results suggest that both dACE2 and hACE2 can support the entry of the pseudovirus bearing the SARS-CoV-2 S into BHK21 cells and HeLa cells as well (Fig. 1b and c). They are consistent with Zhao et al.'s results that dACE2 supports entry into 293T cells by lentiviral particles pseudotyped with SARS-CoV-2 S protein[14].

When infected with the authentic SARS-CoV-2, the number of copies of SARS-CoV-2 *ORF1ab* obviously increased in HeLa cells expressing either dACE2 or hACE2 at 48 and 72 h after infection, compared with that of cells not expressing these two molecules (Fig. 1f). These results confirm that dACE2 is indeed a cellular receptor that supports SARS-CoV-2 infecting host cells, similar to its human ortholog, hACE2.

### The overall structure of dACE2 in complex with RBD. To elaborate the structural basis for dACE2 binding to RBD, we determined the crystal structure of the RBD/dACE2 complex (Supplementary Fig. 2a). The RBD/dACE2 complex was prepared using size exclusion chromatography and the structure was solved to 3.0 Å resolution (Supplementary Table 1), with one RBD binding to a single dACE2 molecule in the asymmetric unit. For dACE2, clear electron densities could be traced for 596 residues

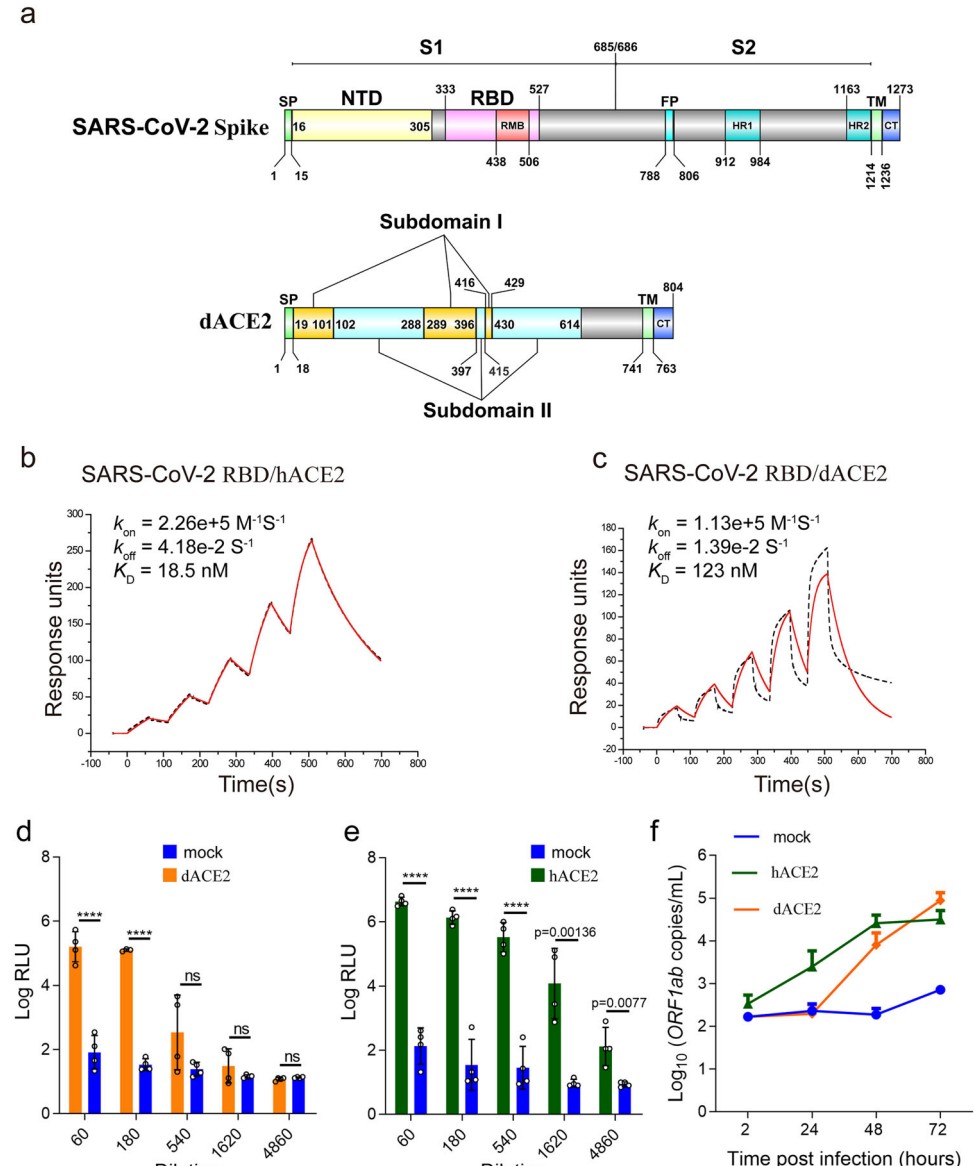

**Fig. 1 The binding affinity of RBD to both hACE2 and dACE2, and the infection efficacies of pseudovirus and authentic virus in hACE2- and dACE2-expressing BHK21 stable cells. a** Schematic of SARS-CoV-2 S (first panel) and dACE2 (second panel) linear compositions colored per domain. **b** SPR-binding curves for immobilized hACE2 with the soluble RBD. **c** SPR-binding curves for immobilized dACE2 with the RBD. The black dashed lines represent the actual data, while the red solid lines represent the fitted results (**b**, **c**). **d** Infection of dACE2-expressing BHK21 cells with SARS-CoV-2 S protein-bearing pseudovirus. **e** Infection of hACE2-expressing BHK21 cells with SARS-CoV-2 S protein-bearing pseudovirus. ****$p < 0.0001$; ns means no significant differences; two-tailed Student's $t$-test. The experiment was performed with quintuplicate cell samples (**d**, **e**). **f** Infection of dACE2/hACE2-expressing HeLa cells with SARS-CoV-2 authentic virus. The experiment was performed with duplicate cell samples. Data are presented as mean values ± SD (**d**-**f**).

from S19 to Y706 and L721 to G725 as well as glycans N-linked to residue N342, while the electron densities for R707 to S720 are invisible. The structure of RBD in the complex includes residues T333 to P526, all of which have a clear density. The overall structure of RBD/dACE2 is very similar to that of the RBD/hACE2 complex (PDB ID: 6LZG) with a root mean square deviation of 0.654 (Supplementary Fig. 2b).

The RBD in the RBD/dACE2 complex structure protein shows the same fold with that in the RBD/hACE2 complex previously reported (PDB ID: 6LZG). It is divided into two subdomains: the β-sheet-dominated conserved core domain, which is stabilized by a disulfide bond between βc2 and βc4, and the loop-dominated external domain, which contains two small β-sheets. The architecture of dACE2 is also similar to that of hACE2 in the RBD/hACE2 complex: it is divided into the N-terminus and Zn²⁺

containing subdomain I, and C-terminus containing subdomain II[15] (Fig. 1a, Supplementary Fig. 2c and d.).

**The interaction interface between dACE2 and RBD and comparison with the RBD/hACE2 complex**. We analyzed the atomic contacts between dACE2 and RBD in the crystal structure of dACE2/RBD with a cutoff distance of 4 Å using CCP4[16]. In the complex, 18 dACE2 residues (Q19, L23, T26, F27, E29, K30, Y33, E36, E37, Y40, Q41, T81, Y82, E325, N329, K352, D354, and R356) form atomic contacts with 18 RBD residues (R403, K417, G446, Y449, Y453, L455, F456, A475, F486, N487, Y489, G496, Q498, T500, N501, G502, Y505, and Q506) (Fig. 2A). The total number of atomic contacts between dACE2 and RBD is 127. An electronic density map of the interaction interface is presented in Supplementary Fig. 3. Among these contacts, 118 are Van der

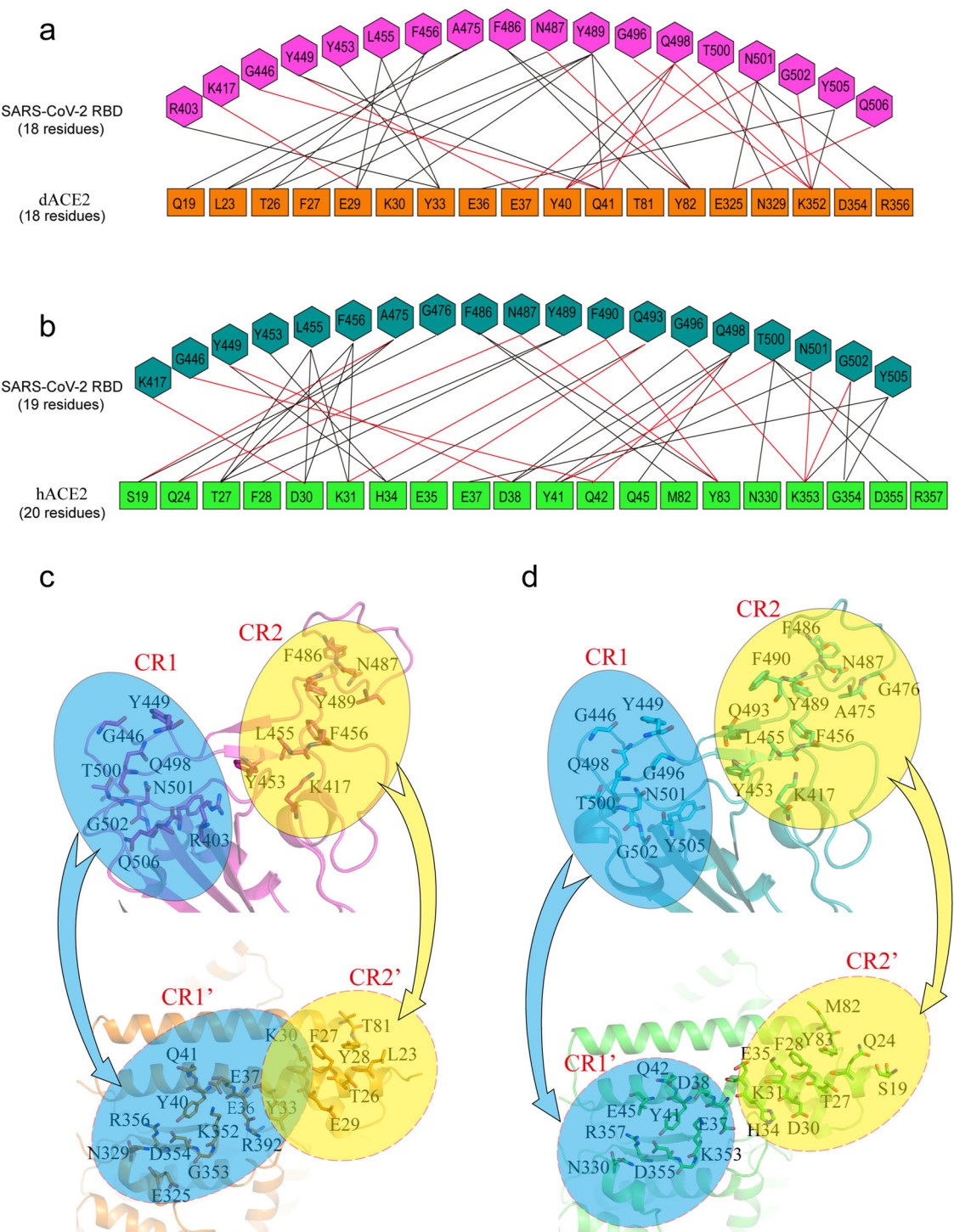

**Fig. 2 Interaction interface in the RBD/dACE2 complex and the RBD/hACE2 complex. a** Residue interactions in the RBD/dACE2 interface. **b** Residue interactions in the RBD/hACE2 (PDB ID: 6LZG) interface. The black lines represent Van der Waals (vdw) interactions, and the red lines represent hydrogen bonds or salt bridges (**a**, **b**). **c** The residues involved in interface interactions in the RBD/dACE2 complex. **d** The residues involved in interface interactions in the RBD/hACE2 complex. The interaction interface in RBD is divided into two contact regions (CR1 and CR2), and their corresponding contact regions in the residues in dACE2 (**c**) or hACE2 (**d**) are named CR1' and CR2', respectively. The residues involved in interface contacts are shown as the sticks representation (**c**, **d**).

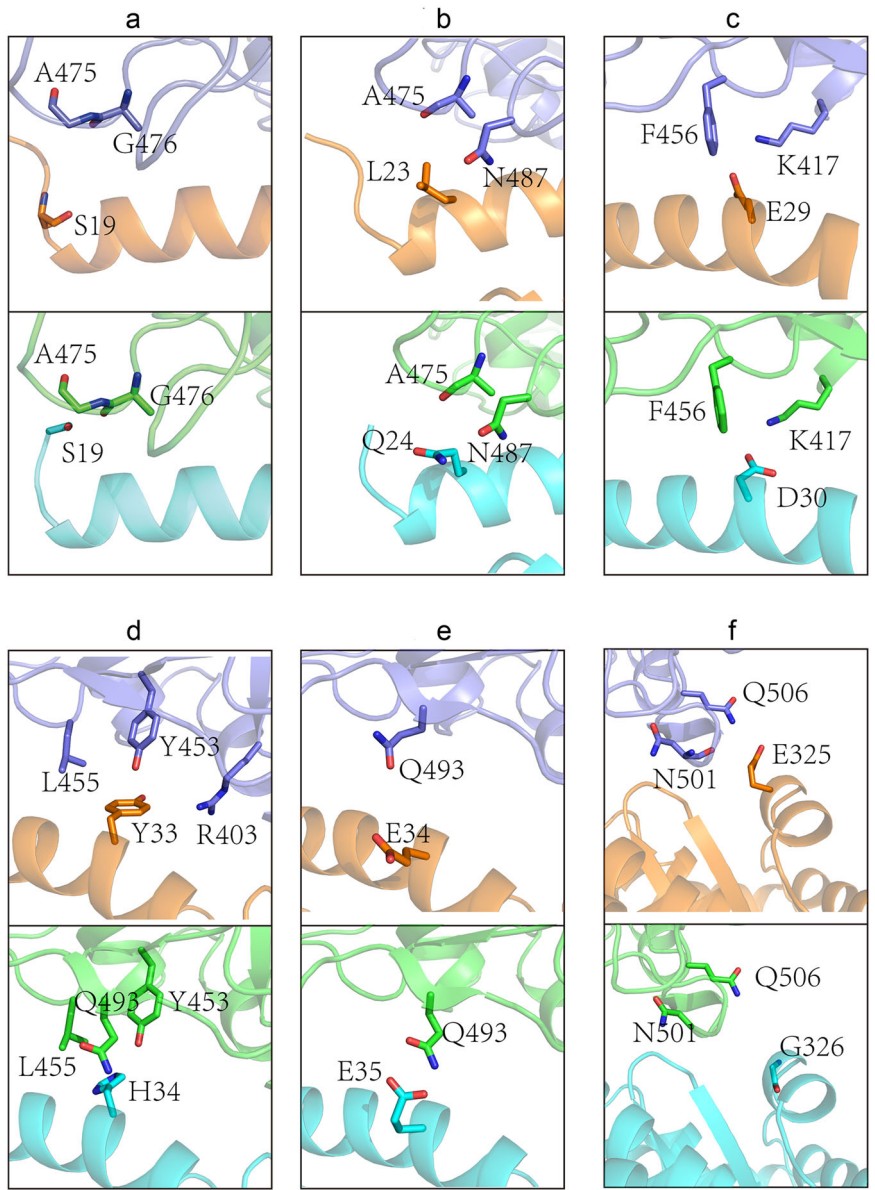

**Fig. 3 Comparison of interface residue contacts at specific positions of dACE2 and hACE2.** In the RBD/dACE2 complex (**a–f** upper panels), RBD residues are colored in light blue, and dACE2 residues are colored in orange. In the RBD/hACE2 complex (**a–f** lower panels), RBD residues are colored in green, and hACE2 residues are colored in cyan.

Waals (vdw) interactions, and 14 are hydrogen bonds or salt bridges (supplementary Tables 2 and 3). The contact interface in the RBD has a saddle shape with two protrusive side parts and a recessed center part, and can be divided into two contact regions (CRs), CR1 (R403, G446, Y449, Q498, T500, N501, G502, and Q506) and CR2 (K417, Y453, L455, F456, F486, N487, and Y489) (Fig. 2C and D). The CR1 is mainly composed of polar residues, whereas the CR2 is mainly composed of hydrophobic and aromatic residues. Generally, RBD CR1 and CR2 interact with two overlapping contact regions in dACE2, CR1' and CR2', respectively (Fig. 2A and C).

In comparison, in the crystal structure of the RBD/hACE2 complex, 20 residues in hACE2 (S19, Q24, T27, F28, D30, K31, H34, E35, E37, D38, Y41, Q42, Q45, M82, Y83, N330, K353, G354, D355, and R357) form atomic contacts with 19 RBD residues (K417, G446, Y449, Y453, L455, F456, A475, G476, F486, N487, Y489, F490, Q493, G496, Q498, T500, N501, G502, and Y505). Of note, the residue T20 in hACE2 is missing in dACE2;

therefor the residue number of dACE2 is one less than that of hACE2 after position 20 (Supplementary Fig. 4). The total number of atomic contacts between RBD and hACE2 is 145, of which 16 are hydrogen bonds or salt bridges (supplementary Table 2 and 3). The contact residues in hACE2 can also be grouped into two contact regions, CR1' and CR2', which are not overlapping (Fig. 2B and D).

We further analyzed the differences in the interface residue contacts at specific positions of dACE2 and hACE2 (Fig. 3). We revealed that dACE2 S19 only makes a vdw contact with SARS-CoV-2 RBD, while hACE2 S19 not only makes vdw contacts with A475 and G476, but also forms a hydrogen bond with A475 (Fig. 3a). Moreover, dACE2 L23 makes three vdw contacts with A475 and N487, but the corresponding hACE2 Q24 forms a hydrogen bond with N487 and 7 vdw contacts with A475 and N487 (Fig. 3b). dACE2 E29 forms a hydrogen bond and a salt bridge with K417, and the corresponding hACE2 D30 forms a hydrogen bond and two salt bridges with K417 (Fig. 3c).

Additionally, dACE2 Y33 interacts with R403, Y453, and L455, whereas the corresponding hACE2 H34 interact with Y453, L455, and Q493 (Fig. 3d). Furthermore, dACE2 E34 does not contact with any SARS-CoV-2 RBD residue, whereas the corresponding hACE2 E35 contacts with Q493 (Fig. 3e). Furthermore, dACE2 E325 interacts with N501 and Q506, whereas the corresponding hACE2 G326 does not contact with any SARS-CoV-2 RBD residue (Fig. 3f).

In summary, slightly fewer residues are involved in forming the interaction interface in the SARS-CoV-2 RBD/dACE2 complex (18 SARS-CoV-2 RBD residues and 18 dACE2 residues) than those in the SARS-CoV-2 RBD/hACE2 complex (19 SARS-CoV-2 RBD residues and 20 dACE2 residues), and the total number of atom contacts, hydrogen bonds and salt bridges in the SARS-CoV-2 RBD/dACE2 complex (127, 13, 1, respectively) are also less than those in the SARS-CoV-2 RBD/hACE2 complex (145, 15, 2, respectively).

**Effect of RBD interface residue mutations on its binding affinity to dACE2 or hACE2.** As mentioned above, at the RBD/dACE2 and RBD/hACE2 interfaces, there is a conserved salt bridge, which is formed between RBD K417 and hACE D30 or dACE E29. Salt bridges are among the strongest non-covalent bonds in protein interface interactions. To address the effect of these salt bridges on the affinity of the binding partners, we introduced K417V or K417N mutations which were found in some SARS-CoV-2 isolates (Supplementary Fig. 5a) to RBD and examined the binding affinity of these mutants to hACE2 and dACE2 using SPR. The results showed that the $K_D$ of RBD with K417V and K417N mutations to dACE2 are 400 and 507 nM, respectively (Fig. 4a and b). Compared with the $K_D$ of the wild type (wt) RBD to dACE2 (123 nM, Fig. 1c), these values represent

3.25- and 4.12-time lower affinities, respectively, suggesting that the salt bridge disruption significantly reduces the affinity of RBD to dACE2. Similarly, the $K_D$ values of RBD with K417V and K417N mutations to hACE2 were calculated to be 53.4 and 49.7 nM, respectively (Fig. 4e and f), which are near three times higher compared to those of the wt RBD to hACE2 (18.5 nM) (Fig. 1b). These results confirm that the disruption of the conserved salt bridge indeed reduces the affinity of RBD to both dACE2 and hACE2.

Of note, the $K_D$ value of the RBD N501Y mutant, which was also detected in SARS-CoV-2 stains (Supplementary Fig. 5b), binding to dACE2 and hACE2 were 37.1 and 0.881 nM (Fig. 4c and g), which are 3.32 and 21.00 times lower than those of wt RBD to dACE2 and hACE2, respectively. Therefore, N501Y mutation enhances the affinity of RBD to both dACE2 and hACE2, among which, the augment is specifically significant for hACE2.

To confirm the effect of RBD mutations on the capacity of binding to native formated ACE2, we measured the binding of RBD mutants to ACE2s expressed on the BHK21 cell surface using flow cytometry (Fig. 4d, h and Supplementary Fig. 6). The results showed that the percentage of the RBD K417N mutant-binding dACE2-positive BHK21 cells was significantly lower than that of the wt RBD-binding dACE2-positive cells. However, the percentage of the RBD N501Y mutant-binding dACE2-positive BHK21 cells was significantly higher than that of the wt RBD-binding dACE2-positive cells. Similarly, among hACE2-positive BHK21 cells, the percentages of both the RBD K417N and K417V mutant-binding cells were significantly lower than that of the wt RBD-binding cells, whereas the percentage of the RBD N501Y mutant-binding cells was significantly higher than that of the wt RBD-binding cells. These results again confirmed the importance

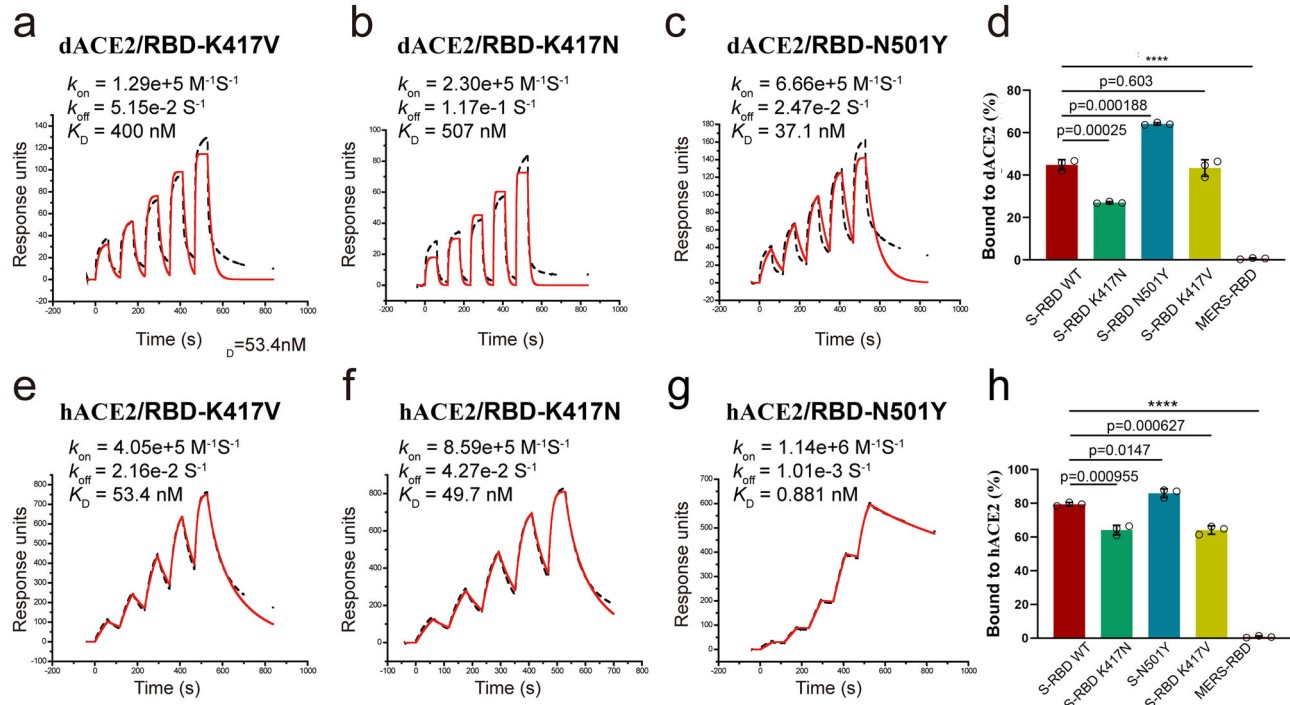

**Fig. 4 The binding ability of RBD mutants to both soluble and transmembrane dACE2 and hACE2. a–c** SPR sensorgram for RBD interface residue mutants binding to dACE2. **e–g** RBD interface residue mutants binding to hACE2. The black dashed lines represent the actual data, while the red solid lines represent the fitted results. The $k_{on}$, $k_{off}$, and $K_D$ values for each mutant are indicated. **d, h** Flow cytometry assay for RBD interface residue mutants binding to dACE2 (**d**) and hACE2 (**h**). The Y-axis represents the percentage of the APC$^+$eGFP$^+$ cells in the eGFP$^+$ cells. Data are presented as mean values ± SD of triplicate cell samples. ****$p < 0.0001$ vs wt; two-tailed Student's t-test.

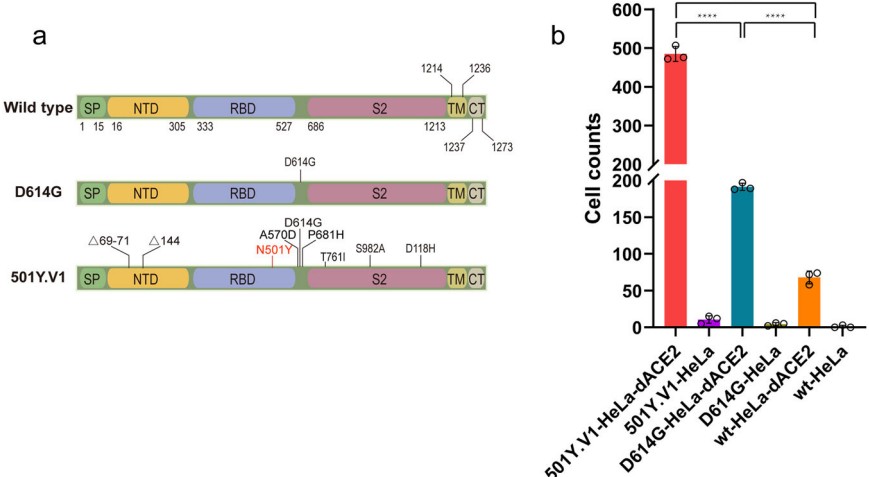

**Fig. 5 D614G, N501Y.V1 variant pseudovirus infection of the dACE2-HeLa cell line. a** Schematic of SARS-CoV-2 S wt and the D614G and N501Y.V1 variants. **b** The dACE2-HeLa cell line was infected with pseudoviruses with SARS-CoV-2 S protein from D614G, N501Y.V1 variants and the wt virus. The number of cells with green fluorescence were counted. Data are presented as mean values ± SD of triplicate cell samples. ****$p < 0.0001$; two-tailed Student's $t$-test.

of RBD interface residues at positions 417 and 501 for determining of the binding affinity to both dACE2 and hACE2 receptors.

**Effect of S protein mutations on pseudotyped SARS-CoV-2 infectivity.** Currently, some major SARS-CoV-2 variants are emerging, including D614G and 501Y.V1 (also referred to B.1.1.7). The D614G variant, having a D614G mutation in the S protein (Fig. 5a), emerged in late January or early February 2020[17]. This variant shows increased infectivity and transmissibility in human respiratory cells and in animal models[18]. SARS-CoV-2 with the D614G mutation has become the dominant form of the virus circulating globally. However, the 501Y.V1 variant is associated with multiple mutations in the S protein, including N501Y, A570D, D614G, P681H, and deletions of H69, V70, and Y144 (Fig. 5a). This variant, transmitting more efficiently than other variants, was first reported on December 14, 2020 in UK, and has been detected in 114 countries until June 23, 2021[19].

To explore the effect of mutations in the S protein of the variants D614G and 501Y.V1 on virus entry into cells expressing dACE2, we constructed three pseudoviruses with S protein from wt SARS-CoV-2, SARS-CoV-2 variants D614G or 501Y.V1, respectively, and tested their infection efficiency. As shown in Fig. 5b, all the three pseudoviruses infected significantly more HeLa cells expressing dACE2 than those not expressing dACE2. In addition, the 501Y.V1 pseudovirus infects significantly more dACE2-expressing cells than both D614G and the wt pseudoviruses. In addition, the D614G pseudovirus infected significantly more dACE2-expressing cells than the wt pseudovirus.

**Discussion**

The finding that SARS-CoV-2 can infect domestic animals has raised a concern that these animals could be a neglected transmission route of this virus[20]. Previous evidence has shown that dogs can be naturally infected with SARS-CoV-2[11,21], and dACE2 can bind to RBD[4]. In the present study, we solved the crystal structure of the RBD/dACE2 complex, and revealed the molecular basis for the recognition of SARS-CoV-2 receptor in dogs. We found that the overall structures of RBD/dACE2 are very similar to the RBD/hACE2 complex. However, the interaction interfaces of the two complexes are slightly different. The number of contact atoms, residues, hydrogen bonds in the RBD/dACE2

interface, are slightly less than those in the RBD/hACE2, which explains the 6.65 times lower affinity of dACE2 for RBD than that of hACE2 (Fig. 1b).

We further showed that naturally occurring interface residue mutations, including K417V, K417N, and N501Y, can significantly modify the affinity of dACE2/hACE2 for RBD. Among them, K417V and K417N, which destroy the sole salt bridge at the interface, reduce the affinity, whereas N501Y increases the affinity. These results confirm the importance of these interface contact residues and validate the interface residue-contact information generated from our crystal structures. Notably, the N501Y mutation not only renders SARS-CoV-2 infectivity in mice, which is not susceptible to wt SARS-CoV-2[22], but also significantly increases the affinity of RBD to dACE2 and hACE2 (Fig. 4). Thus, the N501Y mutation could become a strategy applied by the virus to adapt to various animal species and acquire a wider host range. Actually, N501Y mutation has been detected in SARS-CoV-2 isolated from humans, and the number of virus isolates bearing this mutation continues to increase. As of June 22, 2021, there have been 923,677 N501Y mutation containing strains reported around the world[23]. Hence, this mutation should be closely monitored in naturally circulating strains of SARS-CoV-2.

Apart from the three mutations that we investigated, some other SARS-CoV-2 RDB interface residue mutations that modify the affinity of RBD to hACE2 have recently been reported. Some of them increase the affinity to the hACE2 receptor, such as V367F, W436R, and N354D/D364Y[24]. In addition to N501Y, the naturally occurring N501F and N501T also increased the affinity of RBD to hACE2. There is no evidence showing that they have been selected in the current SARS-CoV-2 pandemic isolates[25]. However, these studies suggest that N501 may be a mutation "hotspot" for the virus to acquire adaptability to the host. These RBD interface residue mutations highlight the necessity to closely monitor virus evolution and to consider them during vaccine development.

In addition to N501 mutations, other mutations, especially D614G, can also increase the infectivity and transmissibility. D614G substitution enhances viral replication by increasing the entry and stability of virions[26]. It disrupts an interprotomer contact in the S protein trimer and veers its conformation toward an ACE2-binding competent state[27]. However, in infected individuals, D614G is associated with elevated upper respiratory tract

viral loads, but not with increased disease severity[27]. In contrast, the N501Y.V1 variant is associated with increased mortality[28]. Therefore, the combination of D614G and N501Y, along with other mutations in the N501Y.V1 variant, may further augment the pathogenicity of the virus. This was confirmed by our data that the N501Y.V1 variant pseudovirus displayed a significantly higher infection efficiency than the D614G variant pseudovirus (Fig. 5).

SARS-CoV-2 can spread from humans to animals, including cats and dogs[29] and spread among cats[30]. Furthermore, this virus can transmit from humans to minks and back to humans[31]. More importantly, the major SARS-CoV-2 variant B.1.1.7 (501Y.V1) has been found in dogs and cats[32]. To date, no evidence has shown that the virus has gained the ability to transmit from cats or dogs to humans. However, our results show that the N501Y mutation can remarkably increase the affinity of RBD to dACE2, and in turn increases the cross-spices transmissibility of the virus. Therefore, monitoring the binding affinity of animal ACE2 to RBD can provide precaution for the occurrence of any new transmission chain and opportunities to nip it in the bud.

## Methods

**Gene expression and protein purification**. The codon-optimized sequence for the ectodomain of dACE2 with a 6×His tag at the C-terminus (all gene sequences for expression of proteins involved in this study are provided in supplementary Table 4) was cloned into the pET21a vector and overexpressed as inclusion bodies in the BL21 (DE3) strain of *Escherichia coli*. Renaturation and purification of dACE2 were performed as previously reported[4,33,34]. Briefly, the dACE2-His inclusion bodies dissolved in dissolution buffer (50 mM Tris-HCl, 100 mM NaCl, 6 M Guanidine hydrochloride, 10% Glycerol, 10 mM EDTA, 10 mM DTT) were injected dropwise and diluted in L-arginine refolding buffer (100 mM Tris-HCl, 400 mM L-Arginine, 2 mM EDTA, 5 mM reduced glutathione, and 0.5 mM oxidized glutathione, pH 8.0). After 24 h, the renatured protein was purified using a Superdex™ 200 Increase 10/300 GL column (GE Healthcare)[35,36] in a gel filtration buffer (20 mM Tris, 150 mM NaCl, pH 8.0). Refolded dACE2-His was used for RBD/dACE2 crystallization and SPR assays.

Moreover, codon-optimized sequence for hACE2 (S19-D615) was fused with mouse Fc and cloned into the pCAGGS vector. To purify hACE2-mFc, the pCAGGS-hACE2-mFc plasmid was transfected into Expi293F cells. After 5 days of expression, the protein was purified using a HiTrap Protein A FF (GE Healthcare) affinity chromatography column in buffer A (20 mM Na₃PO₄, pH 7.4) and buffer B (0.1 M Glycine, pH 3.0) and further purified using Superdex™ 200 Increase 10/300 GL column (GE Healthcare) in a buffer containing 20 mM Na₃PO₄ (pH 7.4). hACE2-mFc was used for SPR assays.

SARS-CoV-2 RBD was expressed as previously reported[4,10,37]. The gene cloned into the Bac-to-Bac baculovirus expression vector was recombined with baculovirus. The recombinant baculovirus was then purified and amplified in sf9 cells, and then used to infect Hi5 cells. The supernatant collected from the cell culture was filtered through a 0.22 μm filter membrane and the protein with a His tag was purified using a HisTrap HP column (GE Healthcare) in buffer A (20 mM Tris, 150 mM NaCl, and pH 8.0) and buffer B (20 mM Tris, 150 mM NaCl, 1 M imidazole, and pH 8.0). The protein was further purified using a Superdex™ 200 Increase 10/300 GL column (GE Healthcare) in a gel filtration buffer (20 mM Tris, 150 mM NaCl, pH 8.0).

**SPR assay**. SPR measurements were performed using a BIAcore 8000 system (GE Healthcare) with CM 5 chips as previously reported (GE Healthcare)[38]. The buffer for all the proteins in the SPR analysis was HBST (20 mM 4-(2-Hydroxyethyl) piperazine-1-ethanesulfonic acid (HEPES), 150 mM NaCl, 0.005% Tween-20, pH 7.4). HBST was used as a running buffer.

A total of 2799 units of hACE2 and 6566 units of dACE2 were immobilized on the CM 5 chip, respectively. RBD was serially diluted (6.25–100 nM or 25–400 nM for hACE2 and dACE2, respectively) and flowed over these two channels. After each cycle, the sensor surface was regenerated using the HBST buffer. The RBD K417N and K417V mutants of serial concentrations from 125–2000 nM flowed over the channel immobilized with 6400 units of dACE2. The RBD K417N and K417V mutants of serial concentrations from 25–400 nM and from 31.25–500 nM, respectively, were flowed over the channel immobilized with 7500 units of hACE2. RBD N501Y was serially diluted to concentrations of 3.125–50 nM or 25–400 nM, and was then flowed over the channels immobilized with 6688 units hACE2 or 5881 units dACE2, respectively. The data were analyzed using the Biacore™ Insight evaluation software (GE healthcare) using a 1:1 Langmuir binding model.

**Pseudovirus infection assay**. Pseudotyped SARS-CoV-2 particles prepared with the vesicular stomatitis virus (VSV) pseudotyped virus packaging system were provided by Weijin Huang from the National Institute for Food and Drug Control. The virus titer was 10^5.8 TCID₅₀/mL as previously reported[39]. The plasmids of full-length hACE2 and dACE2 tagged with eGFP at the C-terminus were transfected into BHK21 cells, respectively. After 24 h, the eGFP-positive cells were sorted using flow cytometry, seeded in 96-well plates (1 × 10^4 cells per well) and then cultured at 5% CO2, and 37 °C for 24 h. Three-fold serial dilutions of the pseudovirus were added to the eGFP-positive cells. After 24 h, the cells were washed with phosphate-buffered saline (PBS) for twice and lysed with the Luciferase Assay System reagent (Promega). Luciferase activity was measured using a GloMax 96 Microplate luminometer (Promega), and the data were analyzed using GraphPad Prism 6.0.

SARS-CoV-2 variant pseudoviruses (501Y.V1 and D614G) were constructed with a GFP-encoding replication-deficient VSV vector backbone (VSV-ΔG-GFP) and the coding sequence of corresponding spike proteins (residues 1–1255)[40,41]. The pseudoviruses were harvested at 20 h post-inoculation. Unpackaged RNA was removed using 0.5 U/μL BaseMuncher endonuclease (Abcam, ab270049) for 1.5 h at 37 °C before pseudoviruses quantification. Viral RNA was extracted (Bioer Technology, Cat# BYQ6.6.101711-213) and quantified using quantitative RT-PCR (qPCR) and a 7500 Fast Real-Time PCR System (Applied Biosystems) with the primers and a probe for detecting the P protein-coding sequence of VSV.

Pseudovirus particles of SARS-CoV-2, 501Y.V1 and D614G were normalized to the same amount before infection. Then, 100 μL of each pseudovirus was added to each well of a 96-well plate containing dACE2-HeLa cells. Untransfected HeLa cells were used as control. After 15 h, the plates were imaged to count the eGFP-positive cells. The number of fluorescent cells was determined using a CQ1 confocal image cytometer (Yokogawa, Japan). Each group contained three replicates.

**Authentic virus infection assay**. HeLa cells overexpressing hACE2 and dACE2 on cell membranes were inoculated with SARS-CoV-2 (hCoV-19/China/CAS-B001/2020, GISAID databases EPI_ISL_514256) at a multiplicity of infection (MOI) of 0.005 and incubated for 1 h at 37 °C. The virus inoculum was then washed twice and replaced with fresh DMEM. Culture supernatants were harvested at 2, 24, 48, and 72 h and then used to extract viral RNA. The viral *ORF1ab* copies were tested using quantitative RT-PCR (forward primer: CCCTGTGGGTTTTACACTTAA; reverse primer: ACGATTGTGCATCAGCTGA, fluorescent probe [P]: 5'- the FAM-CCGTCTGCGGTATGTGGAAAGGTTATGG-BHQ1-3') according to the manual of the One Step PrimeScript RT-PCR kit (Takara, Shiga, Japan).

**Flow cytometry assay**. For flow cytometry analysis, BHK21 cells were transfected with the full-length dACE2 and hACE2 fused with eGFP and incubated in 5% CO2 at 37 °C for 48 h. Then, 2×10^5 cells were resuspended, collected and incubated with RBD or MERS-CoV RBD proteins at a concentration of 5 μg/mL at 37 °C for 30 min. The cells were then washed three times with PBS and stained with anti-His/APC antibody (1:500, Miltenyi Biotec) at 37 °C. After 30 min of incubation, the cells were washed three times with PBS, and analyzed using BD FACSCanto. The assays were independently performed three times. The data were analyzed and visualized with FlowJo software[42].

**Statistics analysis**. The virus infection and flow cytometry data were analyzed using one-way analysis of variance (ANOVA), while the differences between two groups were analyzed using Student's *t*-test. Statistical significance was set at $p < 0.05$.

**Crystallization, data collection, and structure determination**. The sitting-drop method was used to obtain high-resolution crystals[43–45]. In detail, the RBD/dACE2 complex protein was concentrated to 7.5 mg/mL, and 0.8 μL protein was mixed with 0.8 μL reservoir solution. The resulting solution was sealed and equilibrated against 100 μL of the reservoir solution at 18 °C. The high-resolution crystals were grown in 2% 1,4-dioxane, 0.1 M Tris pH 8.0, and 15% polyethylene glycol 3,350.

For data collection, all crystals were cryo-protected by soaking in reservoir solution supplemented with 20% (v/v) glycerol before flash-cooling in liquid nitrogen. Diffraction data were collected at the Shanghai Synchrotron Radiation Facility (SSRF) BL19U. The dataset was processed using HKL2000 software[46]. The structure of the RBD/dACE2 complex was determined using the molecular replacement method and Phaser[47], with previously reported complex structure RBD complex with hACE2 (PDB ID: 6LZG). The atomic models were completed with Coot[48] and refined with phenix.refine in Phenix[47], and the stereochemical qualities of the final models were assessed using MolProbity[49]. The structures were analyzed and visualized with PyMol[50].

**Reporting summary**. Further information on research design is available in the Nature Research Reporting Summary linked to this article.

## Data availability

Atomic coordinates and structure factors have been deposited in the Protein Data Bank under the accession code 7E3J. Other data are available from the corresponding authors upon reasonable request. Source data are provided with this paper.

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

## Acknowledgements

This work was supported by the National Key Research and Development Program of China (2018YFC1200603, 2016YFD0500305, 2020YFC0845900), the Strategic Priority Research Program of CAS (XDB37030204), the National Natural Science Foundation of China (32090014, 31800775), and the National Science and Technology Major Project (2018ZX10733403). G.F.G. is supported by the Yanqi Lake Meeting organized by the Academic Divisions of CAS. We are grateful to the Pathogenic Microbiology and Immunology Public Technology Service Center for its support on flow cytometry assay. We thank Dr. Tong Zhao from the Institutional Center for Shared Technologies and Faculties at the Institute of Microbiology, CAS for assistance with the flow cytometry assay. We also thank the staff of Biosafety Level 3 Laboratory at the Institute of Microbiology, CAS for their help with the authentic virus experiments.

## Author contributions

Y.S., J.Q., H.X., Z.T., and G.F.G. conceived and supervised the study. Z.Z., X.Z., Y.Q.Z., H. Liao, H. Li, and A.Z. prepared protein samples, and carried out SPR and cytometry assay. Z.Z., Y.F.Z., K.L., S.M., and Q.L.W. performed the crystal structure studies. Z.F., L.W., and Q.H.W. did the SPR assay. Y.L., Y.B., Q.L., W.H., and Z.Z. carried out virus assay.

## Competing interests

The authors declare no competing interests.
