## [Peer Review File · Nature Communications]

REVIEWER COMMENTS

Reviewer #1 (Remarks to the Author):

Summary

The ongoing COVID-19 pandemic has resulted in a huge loss of human life and enormous damage to the global economy. Since the emergence of SARS-CoV-2, there has been concern that it could jump from humans into wild or domesticated animals. In this scenario, the virus may lurk in various species, mutate and then resurge in the human population. Indeed, numerous outbreaks in mink farms have been documented and it has been shown that the virus can be transmitted from infected animals back to humans. Therefore, it is of importance to understand the molecular basis for infection of various species by SARS-CoV-2.

The manuscript by Zhang et al demonstrates that that dog angiotensin converting enzyme 2 (dACE2) can bind to SARS-CoV-2 spike protein receptor binding region, and that both pseudotyped and authentic SARS-CoV-2 can infect dACE2-expressing cells. Moreover, they report the crystal structure of the SARS-CoV-2 receptor binding domain (RBD) in complex with dACE2, providing a molecular rationale for the lower binding affinity compared to human ACE2.

The results appear to be scientifically sound, but I think it is fair to say that the manuscript does not offer much in terms of novel biological insight or methodological advancement. Indeed, it has already been shown by Zhao et al that dACE2 can render 293T cells permissive to infection with SARS-CoV-2 S pseudotyped virus. The same study also showed that this is less efficient than human ACE2 (hACE2). This manuscript is not cited in the present study. Regarding the crystal structure, I believe it will be of value for ongoing SARS-CoV-2 research but, in the author's own words, "The overall structure of RBD/dACE2 is very similar to previously reported RBD/hACE2". In addition, a number of studies have performed modelling to predict the interface between the SARS-CoV-2 RBD and ACE2 from various species, including dACE2, so the observations from the crystal structure are unsurprising.

Overall, the present study provides important confirmatory data on an important topic but, given what is already available in the literature, I don't believe this work represents a huge advance.

Suggestions for improvement

- There are numerous issues with spelling and grammar throughout the manuscript which need to be checked carefully. A few examples are listed below:
 - o Replace all instances of "To be noticed" with "Of note".
 - o The title doesn't sound right. I would advise changing it to "Molecular basis for SARS-CoV-2 binding to dog ACE2".
 - o On line 145, "ARS-CoV-2-RBD/hACE2" should be "SARS-CoV-2-RBD/hACE2".
- The description of the interface between dACE2 and the SARS-CoV-2 RBD (lines 154-215) is very monotonous and not suitable to a broad readership. Given that the conclusion is that the interface is very similar to that seen with hACE2, I think this section could be condensed significantly.
- The authors should demonstrate the quality of the electron density for the interfacing

region. This could be included as a supplementary figure or provided to the reviewer.

- Remove “at atomic level” from line 87. You cannot resolve individual atoms at 3 Å resolution, so this is a misleading statement.

References

Zhao X, Chen D, Szabla R, Zheng M, Li G, Du P, Zheng S, Li X, Song C, Li R, Guo JT, Junop M, Zeng H, Lin H. Broad and Differential Animal Angiotensin-Converting Enzyme 2 Receptor Usage by SARS-CoV-2. *J Virol.* 2020 Aug 31;94(18):e00940-20. doi: 10.1128/JVI.00940-20. PMID: 32661139; PMCID: PMC7459545.

Reviewer #2 (Remarks to the Author):

In this work Zhang et al. solved the crystal structure of the SARS-CoV-2 RBD/dACE2 complex, which revealed the molecular basis for SARS-CoV-2 recognized by dACE2. The SPR results indicated the lower binding affinity of RBD/dACE2 than RBD/hACE2, and the residues interaction interface analysis show the numbers of contact atoms are slightly less than those in the RBD/hACE2, which explained the reason for lower binding affinity of RBD/dACE2. Later, the author also measured the binding affinity of natural mutations of SARS-CoV-2 to dACE2 or to hACE2, indicating N501Y mutation facilitates both hACE2 and dACE2 binding affinity and may be a mutation ‘hotspot’ for the virus to acquire adaptability to the host. The work is novel and of great interest to others in receptor recognition of SARS-CoV-2, and provides insight for human-to-animal transmission and potential animal spread of SARS-CoV-2.

Major concerns:

1. For pseudovirus assay in figure 1D and 1E, the result seems corresponding to the conclusion the binding affinity of RBD to dACE2 is lower than that to hACE2, although ACE2 expression level of each cell was not shown. But for the authentic SARS-CoV-2 infection, dACE2-HeLa seems have a higher virus titer on 72 h, not corresponding to the pseudovirus assay. To confirm the cell entry efficiency using different ACE2, the authors are suggested to do experiments in more cell lines
2. Could the author compare the buried surface between RBD/dACE2 and RBD/hACE2?
3. In figure 4, the author measured the different binding affinity of SARS-CoV-2 mutations to dACE2 or hACE2. What’s about the viral entry of these mutations corresponding to figure 4D and 4H?

Minor concerns:

1. Figure 1A lower panel, label the residue number for subdomain boundary.
2. Please indicate the replicates for pseudovirus assay and live virus assay.
3. Please include K_{on} and K_{off} value for each SPR figure.

Reviewer #1 (Remarks to the Author):

...Therefore, it is of importance to understand the molecular basis for infection of various species by SARS-CoV-2...

The results appear to be scientifically sound, but I think it is fair to say that the manuscript does not offer much in terms of novel biological insight or methodological advancement. Indeed, it has already been shown by Zhao et al that dACE2 can render 293T cells permissive to infection with SARS-CoV-2 S pseudotyped virus. The same study also showed that this is less efficient than human ACE2 (hACE2). This manuscript is not cited in the present study.

Response: Thank you for your positive comments and good suggestion. In our study, we performed the SARS-CoV-2 pseudovirus entry assay in both Hela and BHK cell lines with hACE2 or dACE2 expression. Corresponding to Zhao et al.'s work, SARS-CoV-2 pseudovirus enter dACE2 expressing cells less efficiently than hACE2 expressing cells. We have cited Zhao et al.'s work in the revised manuscript. In addition, the similar result was also observed in authentic SARS-CoV-2 virus entry assay in our study.

Regarding the crystal structure, I believe it will be of value for ongoing SARS-CoV-2 research but, in the author's own words, "The overall structure of RBD/dACE2 is very similar to previously reported RBD/hACE2". In addition, a number of studies have performed modelling to predict the interface between the SARS-CoV-2 RBD and ACE2 from various species, including dACE2, so the observations from the crystal structure are unsurprising.

Response: Thank you for this point. The similar overall structure does not mean they have the same binding motif. There are some differences in the interface residue contacts at specific positions of dACE2 and hACE2. For example, the overall structure of the SARS-CoV-2-RBD/hACE2 complex is very similar to the previously reported structure of SARS-RBD bound to the same receptor with an RMSD of 0.431 Å for 669 equivalent Ca atoms (Wang, et al, 2020, Cell. PMID: 32275855), but as we know, there are some key differences on the binding surface which play pivotal role in receptor

binding. In addition, the crystal structure determined experimentally is more accurate than any predicted models, and can serve as an important reference for further work.

Overall, the present study provides important confirmatory data on an important topic but, given what is already available in the literature, I don't believe this work represents a huge advance.

Response: Thank you for your positive comments on the importance of this work. In this study, we determine the mechanism of SARS-CoV-2 RBD in complex with dACE2. We further show that a few important mutations in the RBD binding interface play a pivotal role in the binding affinity of RBD to both dACE2 and hACE2 with our flow cytometry and SPR assays. The pseudovirus entry assay further verified that the N501Y mutation enhanced virus entry efficiency. Our work provides insight for a molecular basis of cross-species transmission and potential animal spread of SARS-CoV-2.

Suggestions for improvement

1. There are numerous issues with spelling and grammar throughout the manuscript which need to be checked carefully.

Response: Thank you for the suggestion. We have had the language polished by a native English speaker.

A few examples are listed below:

(1) Replace all instances of "To be noticed" with "Of note".

Response: Corrected.

(2) The title doesn't sound right. I would advise changing it to "Molecular basis for SARS-CoV-2 binding to dog ACE2".

Response: Corrected.

(3) On line 145, "ARS-CoV-2-RBD/hACE2" should be "SARS-CoV-2-RBD/hACE2".

Response: Corrected.

2. The description of the interface between dACE2 and the SARS-CoV-2 RBD (lines 154-215) is very monotonous and not suitable to a broad readership. Given that the conclusion is that the interface is very similar to that seen with hACE2, I think this section could be condensed significantly.

Response: We have pruned this section to make it more readable.

3. The authors should demonstrate the quality of the electron density for the interfacing region. This could be included as a supplementary figure or provided to the reviewer.

Response: The electron density for the interfacing region is provided as Fig. S3.

4. Remove “at atomic level” from line 87. You cannot resolve individual atoms at 3 Å resolution, so this is a misleading statement.

Response: The “at atomic level” has been deleted.

Reviewer #2 (Remarks to the Author):

In this work Zhang et al. solved the crystal structure of the SARS-CoV-2 RBD/dACE2 complex, which revealed the molecular basis for SARS-CoV-2 recognized by dACE2. The SPR results indicated the lower binding affinity of RBD/dACE2 than RBD/hACE2, and the residues interaction interface analysis show the numbers of contact atoms are slightly less than those in the RBD/hACE2, which explained the reason for lower binding affinity of RBD/dACE2. Later, the author also measured the binding affinity of natural mutations of SARS-CoV-2 to dACE2 or to hACE2, indicating N501Y mutation facilitates both hACE2 and dACE2 binding affinity and may be a mutation ‘hotspot’ for the virus to acquire adaptability to the host. The work is novel and of great interest to others in receptor recognition of SARS-CoV-2, and provides insight for human-to-animal transmission and potential animal spread of SARS-CoV-2.

Response: Thank you for your positive comments on our work.

Major concerns:

1. For pseudovirus assay in figure 1D and 1E, the result seems corresponding to the conclusion the binding affinity of RBD to dACE2 is lower than that to hACE2, although ACE2 expression level of each cell was not shown. But for the authentic SARS-CoV-2 infection, dACE2-HeLa seems have a higher virus titer on 72 h, not corresponding to the pseudovirus assay. To confirm the cell entry efficiency using different ACE2, the authors are suggested to do experiments in more cell lines

Response: It is rather difficult to quantify the absolute ACE2 expression level of each cell. Without the precondition that the dACE2-BHK21 and hACE2-BHK21 cell lines express equal amount of dACE2 and hACE2 in each cell, the comparison of the pseudovirus or authentic virus entry efficiency between the dACE2- and hACE2-expressing cell lines are unwarranted. Therefore, we've deleted the sentence about the comparison of the efficiency of pseudovirus entry induced by dACE2 and hACE2 in the two cell lines. We have performed another experiment to exam the pseudovirus entry capacity in dACE2-Hela and hACE2-Hela cell lines and obtained the similar results. The pseudovirus can infect both of these two cell lines (Fig. S1).

As for the authentic virus infection, there is no statistical difference between dACE2 and hACE2 at 72 h. And Again, without knowing the expression level of each cell, the comparison between virus entry level in these two cell lines makes no sense.

2. Could the author compare the buried surface between RBD/dACE2 and RBD/hACE2?

Response: Thank you for your suggestions. We calculated the buried surface for the RBD/dACE2 and RBD/hACE2 by the Cocomaps server (Vangone et al. Bioinformatics 2011; 27:2915-2916. doi: 10.1093/bioinformatics/btr484), and the results show that the buried surface upon the complex formation for RBD/dACE2 and RBD/hACE2 are 1844.7 Å² and 1755.3 Å², respectively. Although there is a general trend that the affinity increased with the buried area, the binding affinity is not only related to buried surface. The total numbers of Van der Waals' force, hydrogen bonds

and salt bridges at the binding interface in RBD/dACE2 complex are slightly fewer than those in RBD/hACE2 complex, so the binding affinity of SARS-CoV-2 RBD to dACE2 is lower than to hACE2.

3. In figure 4, the author measured the different binding affinity of SARS-CoV-2 mutations to dACE2 or hACE2. What's about the viral entry of these mutations corresponding to figure 4D and 4H?

Response: Thank you for your suggestions. We tested the entry efficiency of two pseudoviruses with S proteins from the two predominant SARS-CoV-2 variants (D614G and N501Y.V1). Our results show that both of these pseudoviruses infect the dACE2-Hela cell line significantly more efficiently than pseudoviruses with wild type S protein. And the pseudovirus with S protein of N501Y.V1 variant, which contains the N501Y mutation, shows significantly higher infection efficiency than that with S protein of D614G variant (Fig 5).

Minor concerns:

1. Figure 1A lower panel, label the residue number for subdomain boundary.

Response: In Figure 1A lower panel, the residue numbers for subdomain boundary have been added.

2. Please indicate the replicates for pseudovirus assay and live virus assay.

Response: The number of replicates (n) have been indicated in the corresponding figure legends.

3. Please include K_{on} and K_{off} value for each SPR figure.

Response: The K_{on} and K_{off} values for each SPR figure have been indicated.

REVIEWER COMMENTS

Reviewer #2 (Remarks to the Author):

My concerns have been addressed. I have no more questions.